# Effect of Pre-Harvest Supplemental UV-A/Blue and Red/Blue LED Lighting on Lettuce Growth and Nutritional Quality

**Triston Hooks [1], Joseph Masabni [2], Ling Sun [1] and Genhua Niu [1,\*]**

1    Texas A&M AgriLife Research Center, 17360 Coit Rd, Dallas, TX 75252, USA;
     triston.hooks@ag.tamu.edu (T.H.); ling.sun@ag.tamu.edu (L.S.)
2    Texas A&M AgriLife Extension, 17360 Coit Rd, Dallas, TX 75252, USA; jmasabni@ag.tamu.edu
\*    Correspondence: gniu@ag.tamu.edu

**Abstract:** Blue light and ultra-violet (UV) light have been shown to influence plant growth, morphology, and quality. In this study, we investigated the effects of pre-harvest supplemental lighting using UV-A and blue (UV-A/Blue) light and red and blue (RB) light on growth and nutritional quality of lettuce grown hydroponically in two greenhouse experiments. The RB spectrum was applied pre-harvest for two days or nights, while the UV-A/Blue spectrum was applied pre-harvest for two or four days or nights. All pre-harvest supplemental lighting treatments had a same duration of 12 h with a photon flux density (PFD) of 171 μmol m$^{-2}$ s$^{-1}$. Results of both experiments showed that pre-harvest supplemental lighting using UV A/Blue or RB light can increase the growth and nutritional quality of lettuce grown hydroponically. The enhancement of lettuce growth and nutritional quality by the pre-harvest supplemental lighting was more effective under low daily light integral (DLI) compared to a high DLI and tended to be more effective when applied during the night, regardless of spectrum.

**Keywords:** anthocyanins; carotenoids; hydroponics; phenolic compounds; phytonutrients; light quality

## 1. Introduction

Greenhouse hydroponic production of leafy greens is expanding globally and is an important component of the world's food supply [1]. Supplemental lighting is commonly used to enhance plant growth and improve quality in greenhouse crop production when natural light is inadequate [2]. For example, leafy green quality, including color and nutritional content, can be improved using commercially available light emitting diodes (LEDs) with customized spectra [3]. However, the electricity cost of supplemental lighting can be significant for a commercial greenhouse, accounting for as much as 30% of the farm gate value [4]. Therefore, there is a need to improve the cost-effectiveness of greenhouse supplemental lighting for long-term sustainable food production.

An alternative to supplemental lighting throughout the production cycle is to incorporate it for only a short period before harvest (pre-harvest) to improve the growth and quality of leafy greens [5]. Applying supplemental lighting when plants are mature with a large canopy may increase light use efficiency. Depending on the duration and efficiency of the supplemental lighting, the consumption of electricity can be significantly reduced [6]. With recent advances in LED efficiency [7] and commercial availability of new spectra (e.g., far-red and ultra-violet), there is a need to evaluate the effectiveness of new LED lights for pre-harvest supplemental lighting on plant growth and quality in greenhouse crop production.

In warmer regions like the Southern United States, greenhouses can benefit from abundant solar radiation and forgo the need to provide supplemental lighting during the summer. However, these greenhouses typically employ various shade materials to regulate temperature by mitigating solar radiation [8]. Aside from reducing light intensity, shade

material can also alter the spectrum of sunlight by attenuating certain wavelengths, particularly in the blue spectrum between 400 and 500 nm [9], which is known to be important in the development of pigments and secondary metabolites in plants [10]. Coincidentally, growers have reported difficulty in developing red and purple pigments in a variety of leafy greens, particularly during summer months when increased shade material is used (personal communication with Jeff Bednar of Profound Microfarms). Additionally, common greenhouse cover material such as glass, polycarbonate, or polyethylene can block up to 95% transmission of ultra-violet (UV) light, while specific UV-absorbing films can block 100% of UV light [11]. Recent studies have shown that UV light, specifically the UV-A spectrum (340–400 nm) may be important for plant growth and quality [12]. Taken together, greenhouse cover and shade materials can significantly alter light intensity and spectrum, specifically in the blue and UV spectra, and this may be an important factor affecting crop growth and quality in greenhouse crop production.

Light spectrum and intensity can influence plant quality and growth. More than three decades ago, McCree [13] observed that red light (600–700 nm) was the most efficient at driving photosynthesis on a quantum yield basis (mole $CO_2$ fixed per mole of absorbed photons). Blue light is less efficient than red light and has been shown to affect plant growth and development by inhibiting cell expansion and division, leading to reduced leaf area [14]. Ideally, blue light can be beneficial to plant growth and photosynthesis when applied in low proportions [15]. Kaiser et al. [16] reported that when low intensity blue light ($<50$ µmol m$^{-2}$ s$^{-1}$) was added to red light in small proportions (6%–12%), then increases in biomass and yield of tomatoes in a greenhouse can be achieved. However, when blue light is added at higher proportions and intensities, it has been shown to increase pigments and secondary metabolites in plants [10]. Zheng et al. [17] showed that 12 h of supplemental blue light at 100 µmol m$^{-2}$ s$^{-1}$ for 10 days before harvest increased antioxidants, like vitamin C and carotenoids, as well as other health-promoting compounds in pak choi (*Brassica campestris* ssp. *chinensis* var. *communis*) plants in a greenhouse. Therefore, blue light should be considered for pre-harvest supplemental lighting to improve crop growth and quality.

Most supplemental lights utilize the visible spectrum between 400 and 700 nm, also known as photosynthetic active radiation (PAR). The PAR spectrum was first understood by Emerson and Lewis [18] due to strong absorption of light by chlorophyll at the 440 nm and 680 nm wavelengths and was later defined by McCree [19]. More recently, however, wavelengths beyond the PAR spectrum have been proven to be beneficial to photosynthesis. For example, Zhen and van Iersel [20] showed that far-red light (700–800 nm) is needed for efficient photochemistry and can lead to increased hypocotyl length, leaf area expansion, and photosynthesis. Similarly, UV light lies outside of the PAR spectrum, but there is growing evidence that the UV-A spectrum (340–400 nm) may be beneficial to plant growth and quality [12].

UV light is a component of the electromagnetic spectrum and constitutes approximately 5% of sunlight when measured at sea level [21]. The UV spectrum ranges from 100 nm to 400 nm and is subdivided into three main regions: UV-A (320–400 nm), UV-B (280–320 nm), and UV-C (100–280 nm). Of the three regions, UV-C radiation has the shortest wavelengths with the greatest ionizing energy that can cause biological damage through inactivation of nucleic acids. Therefore, UV-C radiation has been used in a variety of germicidal and disinfection applications [22]. However, nearly all the UV-C radiation in sunlight is strongly attenuated by ozone in the earth's atmosphere. Therefore, the majority of UV radiation that reaches the earth's surface is in the UV-A and UV-B spectra.

The UV-B spectrum plays an active role in plant physiology by activating the UV-B specific photoreceptor, UV RESISTANT LOCUS8 (UVR8), which has been shown to induce stress responses such as defense signaling and secondary metabolite production in plants [23,24]. Mewis et al. [25] found increased levels of phytonutrients, such as flavonoids, in broccoli sprouts treated with UV-B light from 265–315 nm. However, Wargent et al. [26] found that UV-B light from 280–315 nm reduced leaf area and cellular development in

lettuce plants grown in both field and controlled environment conditions. Balanced doses of UV-B light have been proposed to increase quality of leafy greens and herbs, such as basil, while minimizing reductions in yield [27].

The UV-A spectrum ranges from 340–400 nm and borders the blue spectrum. In plants, UV-A light has been shown to be absorbed by phototropins and cryptochromes, which are the same photoreceptors that capture blue light [28]. Qian et al. [29] reported that UV-A can affect morphogenesis in cucumber plants and leads to robust and dwarfed phenotypes, which can be desirable horticulture traits, with negligible reductions in yield. Brazaityté et al. [30] showed that UV-A can increase pigments and antioxidants in microgreens without reducing growth. Similarly, Moreira-Rodríguez et al. [31] found increased phenolic compounds in broccoli sprouts when treated with UV-A. However, little research has been done on the effect of pre-harvest supplemental lighting containing UV-A spectrum on lettuce in a greenhouse production system.

Therefore, the objective of this study was to evaluate the effect of pre-harvest supplemental lighting using commercially available LED lights with UV and blue wavebands and another common red and blue LED light on the growth and quality of lettuce plants grown in a greenhouse hydroponic system.

## 2. Materials and Methods

### 2.1. Plant Material and Culture

Seeds of leafy lettuce (*Lactuca sativa* L.) var. 'Red Mist' (Osborne Seed Company, Mount Vernon, WA, USA) were sown in 25-mm rockwool cubes (Grodan, Roermond, The Netherlands) in a standard tray (25 × 50 cm) and covered with a dome to maintain moisture and high humidity during germination. The rockwool was pre-rinsed with tap water and pre-soaked with half-strength nutrient solution that had an electrical conductivity (EC) of 1.0 and the pH was adjusted to 6.0 (Table 1). The tray was placed in an indoor propagation rack equipped with air circulation fans and automatic daily sub-irrigation using the same half-strength nutrient solution. After germination in darkness, the dome was removed and three Arize LED light bars (GE Current, Boston, MA, USA) were turned on for 12 h each day for a total photosynthetic photon flux density (PPFD) of 132 $\mu$mol m$^{-2}$ s$^{-1}$. A Watchdog data logger (Spectrum Technologies, Aurora, IL, USA) recorded the average air temperature (23 °C) and relative humidity (45%) in the propagation rack. After seedlings were established with two pairs of true leaves, they were moved to a rooftop greenhouse to harden-off before transplanting.

**Table 1.** Composition of the full-strength, custom nutrient solution used in both experiments.

| Element (mg L$^{-1}$) | | | | | | | | | | | |
|---|---|---|---|---|---|---|---|---|---|---|---|
| N | P | K | Ca | Mg | S | Fe | B | Mn | Zn | Cu | Mo |
| 100 | 20 | 129 | 90 | 26 | 35 | 1.3 | 0.21 | 0.14 | 0.08 | 0.04 | 0.03 |

### 2.2. Rooftop Greenhouse Environment

Two replicated greenhouse experiments were conducted at the Texas A&M AgriLife Research Center in Dallas, TX, USA (32°59′13.2″ N 96°45′59.8″ W; elevation 131 m) from 9/10/20 to 9/24/20 (Experiment 1) and 10/1/20 to 10/15/20 (Experiment 2). The greenhouse temperature was controlled by a heating, ventilation, and air conditioning (HVAC) system equipped in the adjacent office building. Throughout each experiment, greenhouse air temperature, relative humidity, and photosynthetic active radiation (PAR) were recorded by a data logger (Campbell Scientific, Inc., Logan, UT, USA). The actual daily environmental conditions for each experiment are presented in Figure 1.

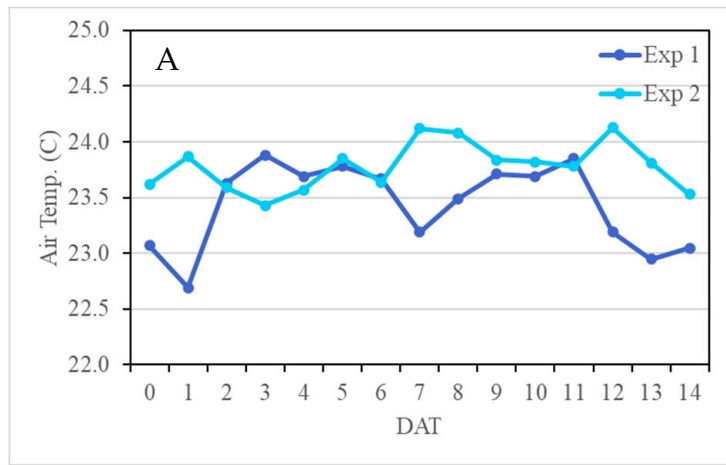

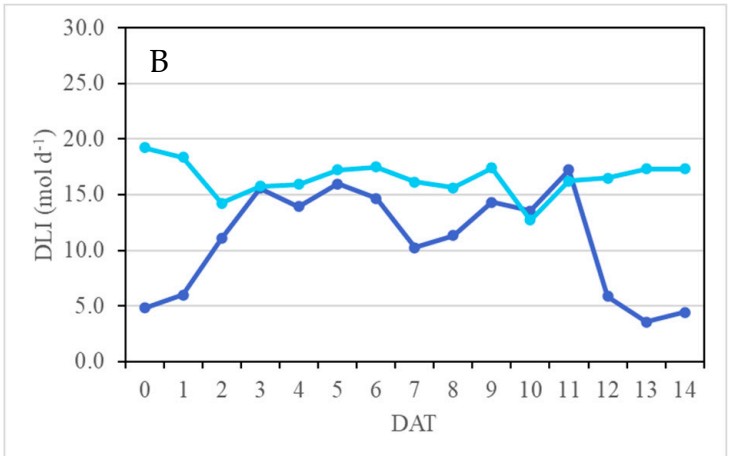

**Figure 1.** Average daily air temperature (**A**) and daily light integral (DLI) (**B**) in the greenhouse throughout the two experiments. The duration of both experiments was 14 days after transplanting (DAT). Experiments 1 and 2 were conducted from 10–24 September and 1–15 October 2020, respectively.

### 2.3. Deep Water Culture Hydroponic System

After seedlings were hardened-off for at least 48 h, they were transplanted into seven deep water culture (DWC) systems for the two-week duration of the experiment. The DWC system consisted of a bin (0.6 m wide × 1.2 m long × 0.2 m deep), a 1.2-m tall frame supporting the bin, and a floating raft (Beaver Plastics, Acheson, AB, Canada) supporting 12 plants that were spaced 20 cm apart. The full-strength custom nutrient solution was prepared with fertilizer salts and tap water and had an EC of 1.5 dS m$^{-1}$ and the pH was adjusted to 6.0 (Table 1). A submersible water pump equipped with a venturi was set up inside the bin to aerate the nutrient solution. A total of seven systems, one for each treatment, were used in both experiments and arranged side-by-side along the West end of the greenhouse for uniform temperature and light distribution.

### 2.4. Pre-Harvest Supplemental Lighting Treatments

There was a total of six pre-harvest supplemental lighting treatments, not including the control (Table 2). The treatments consisted of two spectra: UV-A and blue (UV-A/Blue) and red and blue (RB) LED lights (Fluence by OSRAM, Austin, TX, USA). The UV-A/Blue LED was marketed as UVSpec$^{TM}$ with a peak at 403 nm and comprised of 24% UV-A (between 340 and 400 nm) and 76% blue (between 400 and 500 nm), while the RB LED was marketed as AnthoSpec$^{TM}$ with peaks at 450 and 665 nm and comprised of 56% blue (400–499 nm) and 44% red (600–699 nm), respectively. The spectrum and photon flux densities (PFD) of each treatment were measured at night using a Blue Wave spectroradiometer

(StellarNet, Tampa, FL, USA) and light pollution between each treatment was blocked by hanging reflective insulation material between each system during the pre-harvest treatment duration.

**Table 2.** The six pre-harvest supplemental lighting treatments used in Experiments 1 and 2. The Red and Blue spectrum (Figure A) was 44% red with a peak at 665 nm and 56% blue with a peak at 450 nm. The UV-A/Blue spectrum (Figure B) was 24% UV-A and 76% blue with a peak at 403 nm. All treatments had equal light intensity of 171 μmol m$^{-2}$ s$^{-1}$.

| Treatment ID | Light Spectrum | Duration | |
|--------------|----------------|----------|---|
| RB2D | Red and Blue | Two days * | |
| UV2D | UV-A/Blue | Two days | |
| UV4D | UV-A/Blue | Four days | |
| RB2N | Red and Blue | Two nights | |
| UV2N | UV-A/Blue | Two nights | |
| UV4N | UV-A/Blue | Four nights | |

* Each day or night was a 12-h period from 7 am to 7 pm or 7 pm to 7 am, respectively.

All the light treatments were arranged to have equal PFD of 171 μmol m$^{-2}$ s$^{-1}$. For the UV-A/Blue treatments, two light bars (1.12 m long) were spaced 25 cm apart and hung 20 cm above the plants. For the RB light treatments, one light bar was positioned 46 cm above the plants. The UV-A/Blue treatment was applied for two or four days or nights before harvest (pre-harvest). The RB treatment was applied for a duration of two days or nights pre-harvest, due to limited resources. Digital timers were used to turn the lights on and off during the day (7 a.m. to 7 p.m.) or night (7 p.m. to 7 a.m.). These different time periods were used to evaluate the most effective time to apply the lights treatments.

*2.5. Data Collection*

After two weeks of production in the DWC systems and directly following the pre-harvest supplemental light treatments, nine of the 12 plants were harvested and the following data were collected: shoot fresh weight (FW) and dry weight (DW), shoot water content (WC), total leaf area, specific leaf area (SLA), and relative chlorophyll content (SPAD). Plants were cut at the substrate level and immediately weighed for shoot FW before being placed in paper bags and brought to complete dryness in a drying oven at 60 °C to measure shoot DW using an analytical scale. Shoot WC was calculated as a percentage by taking the difference between shoot FW and DW and dividing by FW. Leaves were separated from the basal stem and the total leaf area per plant was measured using an LI-3100C leaf area meter (LI-COR, Lincoln, NE, USA). The SLA was calculated as the leaf area divided by shoot DW. The SPAD was measured using a handheld SPAD-502Plus meter (Konica Minolta, Osaka, Japan) and was taken as the average of three mature leaves per plant.

Of the remaining three plants per treatment, three representative mature leaves were selected per plant and fresh tissue samples (1 g) were collected and immediately frozen in liquid nitrogen and stored in a freezer at −80 °C. Subsequently, these samples were ground in a mortar and pestle using liquid nitrogen and extracted in methanol. The extracted samples were then analyzed for anthocyanins and total phenolic compounds (TPC) using the methods described by Silva et al. [32] and Ainsworth and Gillespie [33], respectively. Anthocyanins were measured using a Genesys UV-VIS spectrophotometer (Thermo Fisher Scientific, Waltham, MA, USA) and calculated according to the following formula:

$$\frac{V * n * M * A * 100}{\varepsilon * m}$$

where, *V*: The volume of extraction liquid (ml), *n*: Dilution factor, *M*: Molecular weight of cyanidine-3-glucoside (449.2 g), *A*: Absorbance @ 530 nm, *ε*: molar extinction coefficient (29,600), and *m*: weight of sample. TPC was determined using the Folin & Ciocalteu's reagent, measured using a microplate spectrophotometer (BioTek, Winooski, VT, USA) at 765 nm, and reported as mg of Gallic Acid Equivalents (GAE) divided by g of FW.

*2.6. Statistical Analysis*

The seven light treatments were randomized to each of the seven DWC systems and there were nine replications per treatment for all response variables. All data were analyzed as a two-way (experiment and treatment) analysis of variance (ANOVA) using JMP 14 (SAS, Cary, NC, USA). Means were separated using Tukey's honest significant difference (HSD) test at $\alpha = 0.05$.

## 3. Results

*3.1. Experiment 1*

3.1.1. Shoot FW, DW, and WC

All data from both experiments were analyzed together and significant differences between experiments were present for all response variables ($p = 0.0219$ to $< 0.0001$), except for shoot DW ($p = 0.0849$). Because of this, and in addition to the environmental differences (discussed below), the experiments were analyzed and presented separately. There were significant treatment effects in shoot FW, DW, and WC ($p < 0.0005$). Overall, the control tended to have the lowest FW and DW compared to the pre-harvest supplemental light treatments (Figure 2). The shoot FW increased significantly in the RB2D, UV4D, and UV4N treatments (40.5, 40.9, and 48.1 g, respectively) compared to the control (28.5 g), with the greatest increase in the UV4N treatment. The shoot DW followed a similar pattern, and increased significantly in the RB2D, UV4D, RB2N, and UV4N treatments (2.47, 2.49, 2.41, and 2.72 g, respectively) compared to the control (1.65 g), with the greatest increase in the UV4N treatment. Additionally, there were significant increases in FW and DW in UV4N (48.1 and 2.72 g, respectively) compared to UV2N (34.8 and 2.07 g, respectively). Regarding shoot WC, there was a significant reduction only in the RB2N treatment (93.6%) compared to the control (94.2%).

3.1.2. Total Leaf Area, SLA, and SPAD

For total leaf area, SLA, and SPAD, there were significant treatment effects ($p \leq 0.0042$). The total leaf area tended to increase in the pre-harvest supplemental light treatments compared to the control, while SLA behaved in an inverse manner (Figure 3). Total leaf area was significantly greater in the RB2D and UV4N treatments (963 and 956 cm$^2$, respectively) compared to the control (632 cm$^2$), while SLA was significantly reduced in the UV4D and RB2N treatments (33.8 and 33.0 mm$^2$ mgDW$^{-1}$) compared to the control (38.5 mm$^2$ mgDW$^{-1}$). Regarding SPAD, there was a significant reduction in the UV2D treatment compared to the control, RB2D, RB2N, and UV4N treatments.

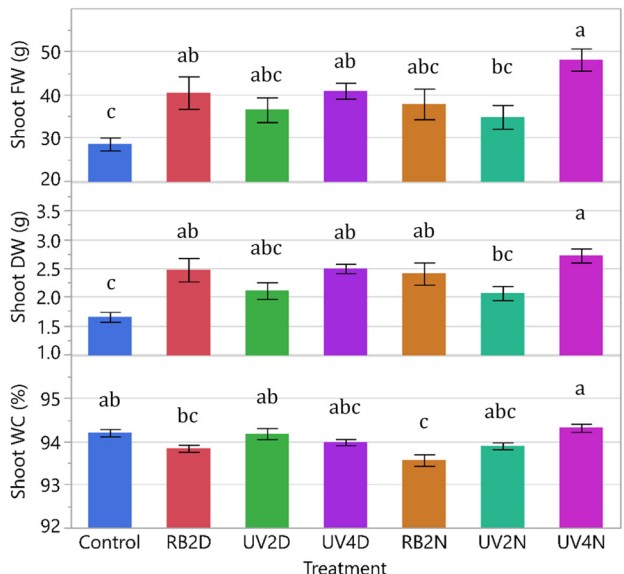

**Figure 2.** Shoot fresh weight (FW), dry weight (DW), and shoot water content (WC) of the lettuce variety 'Red Mist' treated with a pre-harvest supplemental UV-A/Blue (UV) or Red and Blue (RB) LED light for a duration of either two or four days (D) or nights (N) in a greenhouse hydroponic system (Experiment 1). The control plants did not receive any supplemental LED light. See Table 2 for treatment details. Means separated by different letters indicate significant differences according to Tukey's HSD (honestly significant difference) test ($p < 0.05$). Bars represent standard error.

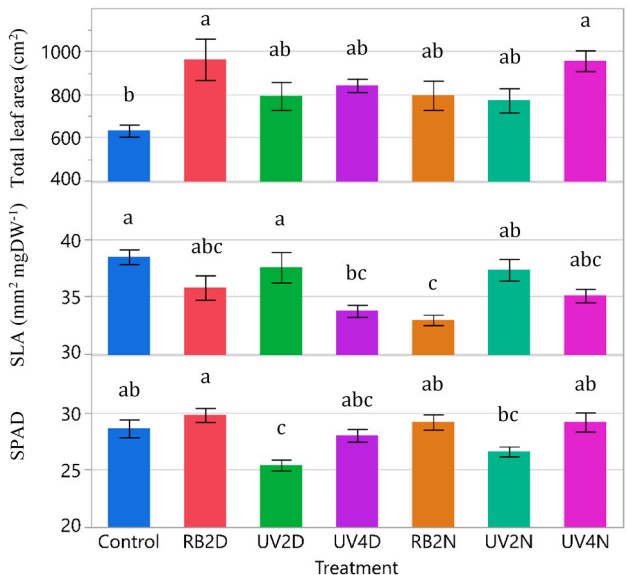

**Figure 3.** Total leaf area, specific leaf area (SLA), and relative chlorophyll content (SPAD), of the lettuce variety 'Red Mist' treated with pre-harvest supplemental UV-A/Blue (UV) or Red and Blue (RB) LED light for a duration of either two or four days (D) or nights (N), in a greenhouse hydroponic system (Experiment 1). The control plants did not receive any supplemental LED light. See Table 2 for treatment details. Means separated by different letters indicate significant differences according to Tukey's HSD (honestly significant difference) test ($p < 0.05$). Bars represent standard error.

### 3.1.3. Phytonutrients

There were significant treatment effects in carotenoids, anthocyanins, and TPC ($p \leq 0.0005$). Overall, phytonutrients tended to increase in the pre-harvest supplemental light treatments compared to the control (Figure 4). The concentration of carotenoids increased in the UV4D, RB2N, and UV2N treatments (277, 261, and 262 μg/gFW) compared to the control

(228 μg/g FW), with the greatest increase in the UV4D treatment. Additionally, UV4D had higher levels of carotenoids compared to UV2D (277 and 244 μg/gFW, respectively). For anthocyanins, there were significant increases in all the treatments compared to the control, with the greatest increase in the RB2N treatment at 218 μg/gFW. For TPC, except for RB2D, there were significant increases in all the treatments compared to the control, with the greatest increase in the RB2N treatment at 7.1 mgGAE/gFW. Notably, the RB2N treatment had higher levels of anthocyanin and TPC (218 μg/gFW and 7.1 mgGAE/gFW, respectively) compared to the RB2D treatment (181 μg/gFW and 4.3 mgGAE/gFW, respectively).

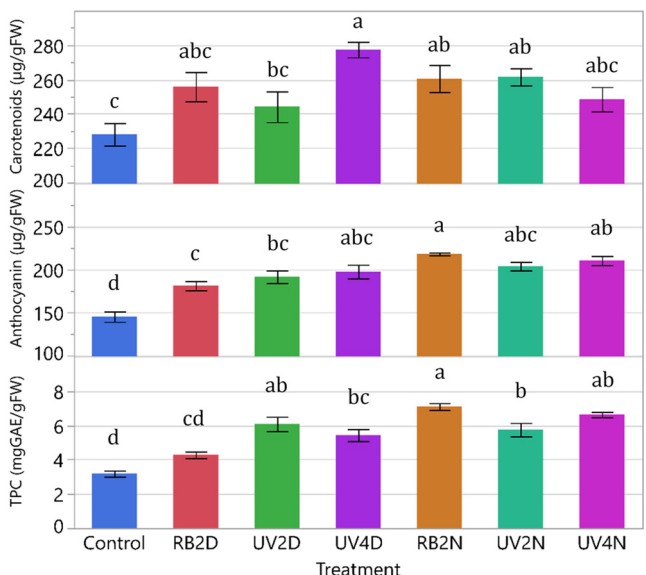

**Figure 4.** Carotenoids, anthocyanin, and total phenolic content (TPC) of leaf tissue samples from the lettuce variety 'Red Mist' treated with pre-harvest supplemental UV-A/Blue (UV) or Red and Blue (RB) LED light for a duration of either two or four days (D) or nights (N), in a greenhouse hydroponic system (Experiment 1). The control plants did not receive any supplemental LED light. See Table 2 for treatment details. Means separated by different letters indicate significant differences according to Tukey's HSD (honestly significant difference) test ($p < 0.05$). Bars represent standard error.

### 3.2. Experiment 2

#### 3.2.1. Shoot FW, DW, and WC

There was a significant treatment effect in shoot DW, but not in shoot FW ($p = 0.0127$ and 0.7045, respectively). For shoot FW, there were marginal differences among the treatments, although all the pre-harvest supplemental light treatments were numerically greater than the control, except for RB2N (Figure 5). For shoot DW, only the UV4N treatment was significantly greater than the control (2.7 and 2.1 g, respectively). Regarding shoot WC, the treatments RB2N, UV2N, and UV4N (94.3%, 94.6%, and 94.4%) were less than the other treatments, including the control (95.3%).

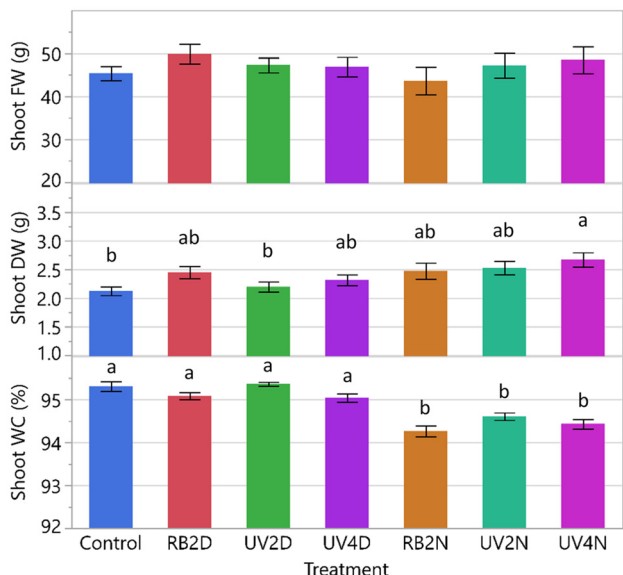

**Figure 5.** Shoot fresh weight (FW), dry weight (DW), and shoot water content (WC) of the lettuce variety 'Red Mist' treated with pre-harvest supplemental UV-A/Blue (UV) or Red and Blue (RB) LED light for a duration of either two or four days (D) or nights (N), in a greenhouse hydroponic system (Experiment 2). The control plants did not receive any supplemental LED light. See Table 2 for treatment details. Means separated by different letters indicate significant differences according to Tukey's HSD (`honestly significant difference`) test ($p < 0.05$). Bars represent standard error.

### 3.2.2. Total Leaf Area, SLA, and SPAD

There were no significant treatment effects for total leaf area ($p = 0.7034$), but there were for SLA and SPAD ($p < 0.0001$). Total leaf area was only marginally reduced, numerically, in the RB2N and UV4N treatments (822 and 873 cm$^2$), compared to the control (877 cm$^2$) (Figure 6). For SLA, there were significant reductions in the RB2N, UV2N, and UV4N treatments (33.1, 36.0, and 32.5 mm$^2$ mgDW$^{-1}$, respectively) compared to the control (41.5 mm$^2$ mgDW$^{-1}$). Regarding SPAD, the same treatments (RB2N, UV2N, and UV4N) were significantly higher (30.3, 27.8, and 28.9, respectively) than the control (23.73).

### 3.2.3. Phytonutrients

There were significant treatment effects in carotenoids, anthocyanins, and TPC ($p = 0.0063$, <0.0001, and <0.0001, respectively). Overall, the control tended to have less phytonutrients in the shoot tissue compared to the pre-harvest supplemental light treatments (Figure 7). The concentration of carotenoids increased significantly only in the UV4N treatment compared to the control (374.2 and 333.7 μg/gFW, respectively). For anthocyanins, there were significant increases in all the treatments compared to the control, except for the UV2D treatment. Additionally, the RB2N treatment increased the most compared to the control (241.5 and 170.6 μg/gFW, respectively). For TPC, there were significant increases only in the UV4D and RB2N treatments (8.0 and 8.5 mgGAE/gFW, respectively) compared to the control (5.9 mgGAE/gFW). Notably, RB2N had significantly greater concentration of anthocyanins and TPC (241.5 μg/gFW and 8.5 mgGAE/gFW, respectively) compared to RB2D (203.9 μg/gFW and 5.5 mgGAE/gFW, respectively) and the UV4D treatment had significantly higher levels of TPC than the UV4N treatment (8.0 and 6.3 mgGAE/gFW, respectively). Moreover, the UV4D treatment was significantly greater than the UV2D treatment in both concentrations of anthocyanin (229.8 and 191.8 μg/gFW, respectively) and TPC (8.0 and 5.8 mgGAE/gFW, respectively).

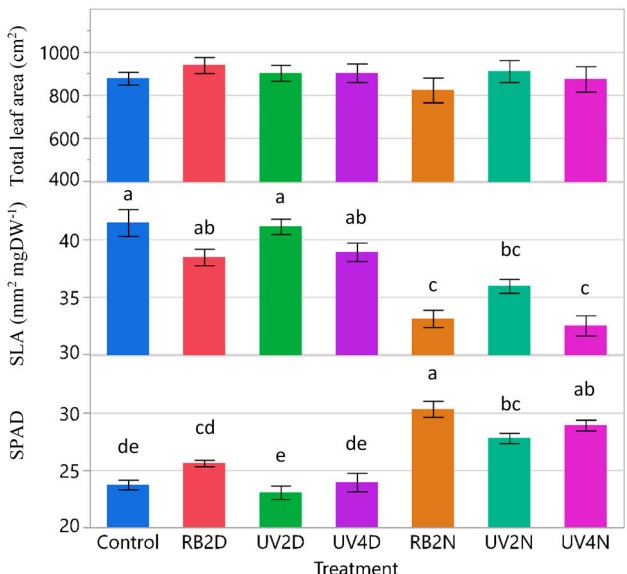

**Figure 6.** Total leaf area, specific leaf area (SLA), and relative chlorophyll content (SPAD), of the lettuce variety 'Red Mist' treated with pre-harvest supplemental UV-A/Blue (UV) or Red and Blue (RB) LED light for a duration of either two or four days (D) or nights (N), in a greenhouse hydroponic system (Experiment 2). The control plants did not receive any supplemental LED light. See Table 2 for treatment details. Means separated by different letters indicate significant differences according to Tukey's HSD (honestly significant difference) test ($p < 0.05$). Bars represent standard error.

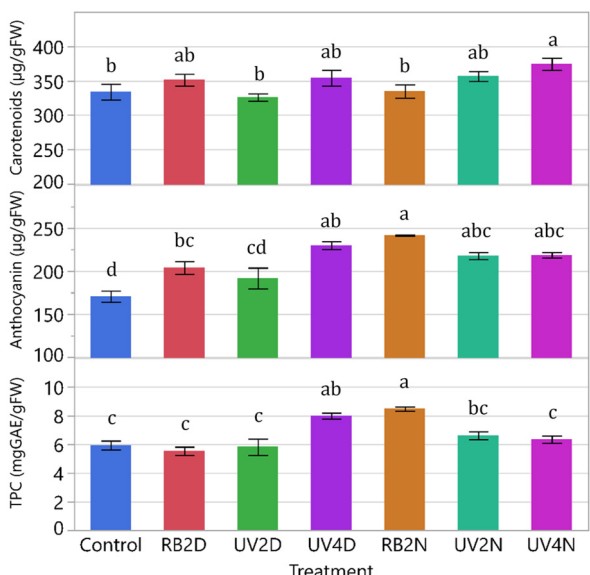

**Figure 7.** Carotenoids, anthocyanin, and total phenolic content (TPC) of leaf tissue samples from the lettuce variety 'Red Mist' treated with pre-harvest supplemental UV-A/Blue (UV) or Red and Blue (RB) LED light for a duration of either two or four days (D) or nights (N), in a greenhouse hydroponic system (Experiment 2). The control plants did not receive any supplemental LED light. See Table 2 for treatment details. Means separated by different letters indicate significant differences according to Tukey's HSD (honestly significant difference) test ($p < 0.05$). Bars represent standard error.

## 4. Discussion

Both experiments were conducted in a similar manner in the greenhouse, with <1 °C difference in the average air temperature between the two experiments. However, there was considerable overcast weather throughout the first experiment compared to the second,

which resulted in an average DLI of 11 mol m$^{-2}$ d$^{-1}$ compared to 17 mol m$^{-2}$ d$^{-1}$ in the second experiment. Lettuce has been shown to increase FW and DW biomass as DLI increases up to 15 mol m$^{-2}$ d$^{-1}$ [34]. As expected, the plants at harvest in the first experiment were on average smaller than plants in the second experiment, particularly in the control treatment where FW and DW were 36% and 28% less, respectively. This led to more pronounced differences between the control and the pre-harvest light treatments in the first experiment compared to the second, which is taken into consideration for the discussion below.

In the first experiment under lower natural DLI, the pre-harvest supplemental light treatments tended to increase shoot FW and DW compared to the control. The UV-A/Blue light increased shoot FW and DW by 28% and 27%, respectively, when applied for four days, and 69% and 65%, respectively, when applied for four nights. Notably, the night treatment appeared to be more effective than the day treatment (although not statistically significant), which is most likely due to the extended photoperiod and consequently increased DLI in the night treatment. Interestingly, this same pattern was not observed in the RB treatments.

In contrast, in the second experiment with a higher natural DLI, the pre-harvest supplemental light treatments did not significantly increase shoot FW and DW compared to the control, except for the UV-A/Blue light when applied for four nights. This indicates that under higher natural DLI, the additional light from the pre-harvest supplemental treatments was not as effective at increasing plant growth compared to lower natural DLI conditions. This may be due to photoinhibition caused by light saturation of photosystem II, which occurs at light intensities above 800 μmol m$^{-2}$ s$^{-1}$ for lettuce [35]. In our study, the average maximum sunlight intensity from both experiments was 950 μmol m$^{-2}$ s$^{-1}$, which typically occurred during midday. With the addition of 171 μmol m$^{-2}$ s$^{-1}$ in the pre-harvest light treatments, the total maximum light intensity was 1121 μmol m$^{-2}$ s$^{-1}$, which is higher than the reported light saturation point for lettuce. Therefore, some photoinhibition occurred during the day when sunlight intensity was at its peak. This could explain the weaker effect of the light treatments on plant growth when applied during the day under higher natural DLI, since the excess photons could not be utilized for biomass production.

In both experiments, there were no significant differences in shoot FW and DW between the RB light or the UV-A/Blue light for the same duration. Even though the RB light treatments contained red light which has the greatest quantum efficiency (mole of photons absorbed per mole of $CO_2$ assimilated) for driving plant growth [36]. Moreover, the UV-A/Blue light provided an overabundance of shortwave light (near 400 nm) compared to longwave light (near 700 nm), that can cause inefficient photosynthesis. Optimal photochemistry for efficient photosynthesis is dependent on the balanced excitation of electrons from photosystem II (PSII) to photosystem I (PSI) [37]. Since PSII has been shown to be more excited by shortwave light [38] and PSI has been shown to be more excited by longwave light, such as far-red [20], then an overabundance of UV-A/Blue light can cause an over-excitation in PSII and an under-excitation in PSI. This would result in inefficient photosynthesis in the UV-A/Blue light compared to the RB light; however, the subsequent effect on plant growth was not observed in this study. This may be due to the lack of far-red light in the treatments, which has been shown to enhance photosynthesis [20].

In Experiment 2, shoot WC was reduced by an average of 1% when the pre-harvest light treatments were applied at night compared to during the day and the control. Since a lower shoot WC indicates greater dry biomass, then the pre-harvest supplemental lighting was more effective at increasing dry biomass when applied at night compared to during the day, regardless of spectrum (UV-A/Blue or RB). This was consistent with the trends in shoot FW and DW which were noted above.

The total leaf area followed similar patterns as shoot FW in both experiments. However, SLA tended to decrease in the pre-harvest supplemental light treatments, particularly when applied at night. Notably, in Experiment 2 when the natural DLI was higher, SLA was reduced by an average of 18% compared to the control when UV-A/Blue or RB light

treatments were applied at night. This indicates that the leaves were thicker and more compact in these treatments. This was further supported by the SPAD data in Experiment 2, where there was an average increase of 22% in the UV-A/Blue and RB treatments at night, compared to the control. SPAD is an indicator of the "greenness" or relative chlorophyll content in the leaves and is used as an indicator of plant health [39], but has also been associated with darker colored and thicker leaves [27]. Taken together, these results indicate that UV-A/Blue and RB light can influence plant morphology by increasing leaf dry matter and thickness and reducing leaf area. This is consistent with previous findings that high levels of blue light can reduce leaf area [40]. In addition to blue light, our results also show that UV-A/Blue light with a peak of 403 nm has a similar effect on leaf morphology.

Secondary metabolites, also called phytonutrients, comprise a large family of plant compounds that are important sources of antioxidants. Antioxidants have been shown to be increasingly important in our diet and health by actively scavenging free radicals, reducing inflammation, and lowering risks to certain diseases [41,42]. In the current study, both experiments showed that the UV-A/Blue and RB light treatments increased levels of phytonutrients, including carotenoids, anthocyanins, and TPC compared to the control. Regarding the UV-A/Blue light, our results are consistent with those of Brazaityte et al. [43], who used a similar UV-A light with a peak wavelength of 402 nm and reported high levels of TPC in mustard microgreens.

Carotenoids and anthocyanins are natural pigments in plants with antioxidant properties [44]; therefore, plants would be expected to develop more coloration under the UV-A/Blue and RB treatments. Based on the visual appearance of the plants at the time of harvest, this was consistent with the data (Supplemental Figure S1). Notably, the RB treatment when applied at night led to the greatest increase in anthocyanins, with an average increase of 46% compared to the control.

When the UV-A/Blue light was applied for a duration of only two days or nights, carotenoids, anthocyanins, and TPC increased by an average of 11%, 36%, and 86%, respectively, compared to the control, although this was only found in the first experiment under lower natural DLI. In the second experiment, carotenoids, anthocyanins, and TPC increased by an average of 9%, 31%, and 21%, respectively, compared to the control, but only when the UV-A/Blue light was applied for four days or nights. This indicates the effect of pre-harvest supplemental lighting with UV-A/Blue light is influenced by the "background" DLI; the effect was more significant at a lower "background" DLI. Dou et al. [27] reported that the pre-harvest UV-B light treatment had a bigger increase in concentrations of anthocyanin, phenolics, and flavonoids in basil (*Ocimum basilicum*) under a lower PPFD compared to higher PPFD.

## 5. Conclusions

Overall, both experiments showed that pre-harvest supplemental light treatments using UV-A/Blue light or RB light can increase lettuce growth and quality in a greenhouse hydroponic system. Additionally, considering the increases in shoot DW and reductions in shoot WC, the pre-harvest supplemental lighting during the night were more effective at increasing plant biomass, compared to during the day. The results in this study also showed that UV-A/Blue or RB light can change leaf morphology by making leaves thicker and more compact. In regard to phytonutrient content, our results showed that UV-A/Blue or RB light can similarly increase beneficial antioxidants such as carotenoids, anthocyanins, and phenolic compounds in lettuce. Furthermore, the increase in plant quality was more pronounced under lower natural DLI conditions. More research is needed to further understand the effects of pre-harvest supplemental lighting spectrum and intensity on plant growth and quality in different growing seasons due to the nature of uncontrollable DLI of sunlight.

**Supplementary Materials:** The following are available online at https://www.mdpi.com/article/10 .3390/horticulturae7040080/s1, Figure S1: Representative photos of the lettuce variety 'Red Mist' treated with supplemental UV-A/Blue (UV) or Red and Blue (RB) LED light for a duration of either

two or four days (D) or nights (N) pre-harvest, in a greenhouse hydroponic system. The control plants did not receive any supplemental LED light. See Table 2 for additional treatment details

**Author Contributions:** Conceptualization, G.N. and T.H.; methodology, T.H., G.N., J.M., and L.S.; validation, T. Triston and G.N.; formal analysis, T.H. and L.S.; investigation, T.H. and L.S.; data curation, T.H. and L.S.; writing—original draft preparation, T.H.; writing—review and editing, G.N. and J.M.; supervision, G.N.; project administration, G.N.; funding acquisition, G.N. All authors have read and agreed to the published version of the manuscript.

**Funding:** This research is partially supported by Texas A&M AgriLife Research and USDA Hatch project TEX07726.

**Acknowledgments:** The authors appreciate the donation of LED lights from Fluence Bioengineering, Austin, TX. Mention of a trademark, proprietary product, or vendor does not constitute a guarantee or warranty of the product by Texas A&M AgriLife Research or USDA and does not imply its approval of exclusion of other products or vendors that may be suitable.

**Conflicts of Interest:** The authors declare no conflict of interest.

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
