# Peer review of "Effect of Pre-Harvest Supplemental UV-A/Blue and Red/Blue LED Lighting on Lettuce Growth and Nutritional Quality"

_horticulturae, doi:10.3390/horticulturae7040080_

Round 1

Reviewer 1 Report

The manuscript entitled “Effect of pre-harvest supplemental UV-A/Blue and Red/Blue LED lighting on lettuce growth and nutritional quality” analyzed the effect of pre-harvest supplemental lighting using different (UV-A/blue or red/blue) commercially available LED lights on the quality and growth of lettuce var. Red Mist. The supplemental light was provided to plants for 2 or 4 days (7 am – 7 pm) or 2 or 4 nights (7 pm – 7 am) during two experiments carried out during September 2020. Authors showed that the supplemental light was more effective when applied during the night than when applied during the day.

The topic of the work resulted interesting. However, some revisions must be done:

Abstract

The Abstract results well written.

Introduction

Information about the topic result completed and well described.

Materials and Methods

The experiments are well described and full of details.

Figure 1: the date of experiments results different than that reported at lines 137-138.

Figure 1: remove the two full stops at the end of the caption of the table.

Table 2: In the caption of the table –1 and –2 must be superscripted. The first letter of the words “Experiments” and “Figure” must be written in capital letters.

Results

Lines 224-227: the significant differences between the experiments mean that a third experiment is necessary or however another season experiment because many significant differences between experiments can’t lead to a general concept or theory.

Figure 3 and 6: the measurement units in Figure result different written when compared with the measurement unit written in the text.

Lines 301-303 and 354-356: since the results are not significant, I think that these sentences are unnecessary.

Line 379 “and subsequently ferrodoxin”: it is not clear this affirmation by authors. Ferrodoxin is ferredoxin.

Line 384 “the subsequent effect on plant growth was not observed in the study”: this sentence must be discussed. Why was the expected effect not observed? Could some photoinhibition regulation sentence such as spillover or xanthophyll cycle be present?

Author Response

Response to Reviewer #1 Comments

  1. Figure 1: the date of experiments results different than that reported at lines 137-138.

Response: Corrected in Figure 1.

  1. Figure 1: remove the two full stops at the end of the caption of the table.

Response: Corrected.

  1. Table 2: In the caption of the table –1 and –2 must be superscripted. The first letter of the words “Experiments” and “Figure” must be written in capital letters.

Response: Corrected.

  1. Lines 224-227: the significant differences between the experiments mean that a third experiment is necessary or however another season experiment because many significant differences between experiments can’t lead to a general concept or theory.

Response: The differences between the experiments were due to documented environmental conditions (lower natural DLI in Experiment 1 due to persistent overcast weather) that was openly explained and discussed in the paper (see Discussion lines 337 – 346, 355 – 360, 435-427). Due to these differences, we presented the results of the two experiments separately and drew conclusions based on consistent results in both experiments (e.g. increased shoot DW and phytonutrients in the pre-harvest light treatments compared to the control). Additionally, we observed the background DLI effect which was an interesting discussion point (see lines 425 – 430). We believe the two replicated experiments that we conducted provide a combination of both consistency and uniqueness to offer sufficient conclusions and room for further research.

  1. Figure 3 and 6: the measurement units in Figure result different written when compared with the measurement unit written in the text.

Response: Values in the text and Figures were double checked and were confirmed to be correct. However, some of the mean separation letters in the figures were covered due to formatting issues and this was corrected and now matches what is written in the text.

  1. Lines 301-303 and 354-356: since the results are not significant, I think that these sentences are unnecessary.

Response: This is correct and can be removed if insisted, but we would like to leave these sentences in because they are the only mention of the results of those response variables.

  1. Line 379 “and subsequently ferrodoxin”: it is not clear this affirmation by authors. Ferrodoxin is ferredoxin.

Response: We agree and removed the text about ferrodoxin from the sentence so as not to cause confusion.

  1. Line 384 “the subsequent effect on plant growth was not observed in the study”: this sentence must be discussed. Why was the expected effect not observed? Could some photoinhibition regulation sentence such as spillover or xanthophyll cycle be present?

Response: An additional sentence was added to provide a reason as to why this was observed. We believe it is related to the efficiency of photosynthesis between PSII and PSI as discussed in the same paragraph. Photoinhibition is certainly occurring to some extent and is discussed in the following paragraphs. The xanthophyll cycle is a very interesting point although we don’t feel equipped to discuss this point due to the fact that xanthophylls were not measured in this study. We will keep this in mind for future research.

Reviewer 2 Report

 It is very interesting study. Usually, those types of the study are performed in plant factory with artificial lighting. But this study has been performed in greenhouse. It may be more practical for commercial vegetable production.

 I have some questions and comments as written below. And did Ca deficit symptom occur in the experiments?

  1. 116: I did not understand the meaning of ‘UV-A/blue and red/blue’ because this is the first appearance. Please explain more.

  1. 124: Was the composition of full-strength solution described in the previous paper? If so, please cite it.

  1. 139: First appearance of ‘HVAC’. ‘Heating Ventilation and Air Conditioning (HVAC)’ may be better.

Figure 1 (and others): Do the black bars indicate standard deviation, or standard error? Please add the explanation as a footnote.

Author Response

Response to Reviewer #2 Comments

  1. It is very interesting study. Usually, those types of the study are performed in plant factory with artificial lighting. But this study has been performed in greenhouse. It may be more practical for commercial vegetable production.

It is a common practice using supplemental lighting in greenhouses in winter season when the natural light is low, but this would increase the production costs (lights and electricity). Nowadays, there are many new LED lights with different spectrums. But not enough studies have been carried out to quantify the effectiveness. Also, our study is pre-harvest or end-of-production (EOP) light treatment.

  1. I have some questions and comments as written below. And did Ca deficit symptom occur in the experiments?

Response: We are confident that Ca deficient symptoms did not occur in the plants in either of the experiments due to high levels of Ca in our tap water (~60 ppm) which when added to the amount in the full-strength nutrient solution would equal 150 ppm which is sufficient for leafy greens. Also, the greenhouse is well ventilated. Additionally, the volume of the hydroponic bins was ~25 gallons per 12 plants which is about 2 gal per plant which is a sufficient buffer for one growth cycle. Overall, no nutrient deficiencies of any kind were observed in any of the plants during both of the experiments.

  1. 116: I did not understand the meaning of ‘UV-A/blue and red/blue’ because this is the first appearance. Please explain more.

Response: This is a good point and the text was revised to indicate that commercially available LED lights with UV and blue wavebands and another common red and blue LED light were used. The detailed descriptions of LED spectrum remain defined in the M&M section.

  1. 124: Was the composition of full-strength solution described in the previous paper? If so, please cite it.

Response: The composition of the full-strength nutrient solution was custom and prepared with fertilizer salts with tap water, which was added to the text for better clarification. The solution was not referenced in another paper.

  1. 139: First appearance of ‘HVAC’. ‘Heating Ventilation and Air Conditioning (HVAC)’ may be better.

Response: Corrected as indicated.

  1. Figure 1 (and others): Do the black bars indicate standard deviation, or standard error? Please add the explanation as a footnote.

Response: The black bars represent standard errors, and this was added to each figure as indicated.

Round 2

Reviewer 1 Report

I appreciate the responses by authors to my comments. They correctly modified some underlined mistakes.

Neverthless, I disagree about the significant variability between the experiments due to the environment conditions. Probably, they could conclude that further researches are necessary to verify the effectivness of the use of different lights maintaining constant environment conditions. 

Moreover, they didn't format references for Horticulturae.

Author Response

Thank you again for reviewing our revised version. We have revised the conclusion to explicitly say that more research is needed in different seasons due to the fact of uncontrollable DLI of sunlight. 

We have revised the format of references.
